# Selective Removal of the Emerging Dye Basic Blue 3 via Molecularly Imprinting Technique

**DOI:** 10.3390/molecules27103276

**Published:** 2022-05-19

**Authors:** Maria Sadia, Izaz Ahmed, Faiz Ali, Muhammad Zahoor, Riaz Ullah, Farhat Ali Khan, Essam A. Ali, Amir Sohail

**Affiliations:** 1Department of Chemistry, University of Malakand, Chakdara, Lower Dir 18800, Pakistan; izazahmadij@gmail.com (I.A.); faizy186@gmail.com (F.A.); 2Department of Biochemistry, University of Malakand, Chakdara, Lower Dir 18800, Pakistan; 3Department of Pharmacognosy, College of Pharmacy, King Saud University, Riyadh 11451, Saudi Arabia; rullah@ksu.edu.sa; 4Department of Pharmacy, Shaheed Benazir Bhutto University, Sheringal, Dir Upper 18000, Pakistan; farhatkhan2k9@yahoo.com; 5Department of Pharmaceutical Chemistry, College of Pharmacy, King Saud University, Riyadh 11451, Saudi Arabia; esali@ksu.edu.sa; 6MSC Construction Project Management, University of Bolton, Bolton BL3 5AB, UK; syedaamirsohail89@gmail.com

**Keywords:** molecularly imprinting polymer, adsorption, Basic Blue, environmental pollution

## Abstract

A molecularly imprinting polymer (MIP) was synthesized for Basic Blue 3 dye and applied to wastewater for the adsorption of a target template. The MIPs were synthesized by bulk polymerization using methacrylic acid (MAA) and ethylene glycol dimethacrylate (EGDMA). Basic Blue 3 dye (BB-3), 2,2′-azobisisobutyronitrile (AIBN) and methanol were used as a functional monomer, cross linker, template, initiator and porogenic solvent, respectively, while non-imprinting polymers (NIP) were synthesized by the same procedure but without template molecules. The contact time was 25 min for the adsorption of BB-3 dye from 10 mL of spiked solution using 25 mg polymer. The adsorption of dye (BB-3) on the MIP followed the pseudo-second order kinetic (k_2_ = 0.0079 mg·g^−1^·min^−1^), and it was according to the Langmuir isotherm, with maximum adsorption capacities of 78.13, 85.4 and 99.0 mg·g^−1^ of the MIP at 283 K, 298 K and 313 K, respectively and 7 mg·g^−1^ for the NIP. The negative values of ΔG° indicate that the removal of dye by the molecularly imprinting polymer and non-imprinting polymer is spontaneous, and the positive values of ΔH° and ΔS° indicate that the process is endothermic and occurred with the increase of randomness. The selectivity of the MIP for BB-3 dye was investigated in the presence of structurally similar as well as different dyes, but the MIP showed higher selectivity than the NIP. The imprinted polymer showed 96% rebinding capacity at 313 K towards the template, and the calculated imprinted factor and Kd value were 10.73 and 2.62, respectively. In this work, the MIP showed a greater potential of selectivity for the template from wastewater relative to the closely similar compounds.

## 1. Introduction

Environmental pollution caused by wastewater discharging from industries containing dyes is a worldwide problem. About 7 × 10^5^–1 × 10^6^ tons of synthetic dyes are produced annually around the world [1]. In these dyes, nearly 50% are discarded during the coloring process, and approximately 10–15% are lost as effluent into the environment. Synthetic and natural dyes are mostly used in industries such as food, pharmaceuticals, leathers, pulps, plastics, cosmetics, textiles, etc. [2]. When the effluents of these industries release into the river; they cause the degradation of that environment, such as changing the natural color and the formation of foam on the water surface. Synthetic dyes are generally toxic, mutagenic, carcinogenic and lethal for the health of different organisms (algae and phytoplankton) [3]. Dye concentration of even less than 1 mg/L causes intense water coloration. It can reduce the penetration of oxygen and light to the aquatic environment and result in a decrease in photosynthetic activity.

The common wastewater plant treatment is not efficient in removing toxic compounds such as dyes from water [4]. For the removal and determination of these toxic dyes from industrial effluent, different physical and chemical techniques are used such as chemical oxidation, chemical precipitation, adsorption, photocatalysis, microbial or enzymatic treatment, natural sorbent materials [5], mineral composite, chemical oxidation, coagulation–flocculation, irradiation, adsorption, precipitation, membrane technologies, a combination of an aerobic and anaerobic and ion exchange processes [6]. Therefore, all of these techniques have some drawbacks and problems such as high cost, large solvent consumption, large analysis time and tough sample preparation, etc. Among these techniques, adsorptions using solid adsorbent are effective because of the simplicity and ease of their operation. However, simple solid sorbent also has some limitations such as high cost and lack of selectivity [7]. The biological degradation method is also used but cannot eliminate the color of many synthetic dyes completely because of their biodegradable difficulty. To overcome the above limitation, more specific and selective methods are needed to remove pollutants from environmental water. So, MIP is the best choice because of its specificity and selectivity [8]. Among the published work, MIPs for dyes are used as a sorbent for the solid phase extraction because of their high selectivity, even in complex samples [9].

Molecularly imprinting polymers (MIPs) are the synthetic polymer used for the recognition of specific substances and are able to bind with target molecules. MIPs can be synthesized by mixing template (analyte) and functional monomer to form pure complex. Then, a cross linker is added to stabilize the MIP’s 3-dimensional structure [10]. After the polymerization, the template is removed by washing, creating cavities which are complementary to the template’s shape, size and functional group position.

Most commonly, MIPs interact with target molecules through ionic interaction, hydrogen interaction or non-covalent bonding [11]. The successful synthesis of MIP depends on the cross linker, monomer and the appropriate condition of polymerization for the target analyte. There are different methods for MIP preparation such as suspension polymerization, bulk polymerization, precipitation polymerization and two-step polymerization. The preparation of MIP through Bulk polymerization is the simplest method to form a large-sized monolith, which can be sieved and ground by mortar and pestle into a smaller size of 5–50 µm of irregular shape particles [12]. These synthetic, tailor-made receptors have some advantages over the recognition of biological substances such as antibodies and enzymes as they are cheaper, easily prepared and resistant to harsh conditions such as base, acid, high temperature, organic solvents and high pressure. In contrast to biological molecules, they are less expensive as well [13].

MIPs have been applied successfully to remove pesticides, pharmaceutical products and products of personal care as sensors and in solid phase extraction (SPE). MIPs and non-imprinted polymers (NIPs) can also be used in many areas such as control drug delivery, biomedical or analytical diagnosis, bio sensor SPE and chromatography, as well as for viruses and toxins [14].

The main goal of this research work is to design a highly selective MIP sorbent for Basic Blue 3 dye and its selectivity, rebinding and application in different effluents. For the fast determination of Basic Blue 3 dye in water samples, MIPs and NIPs were synthesized through bulk polymerization, in which Basic Blue 3 dye shows more selectively toward MIPs as compared to NIPs because of the recognition property of the MIP network.

## 2. Materials and Methods

### 2.1. Materials

All solvents and dyes were of analytical grade and were supplied from Sigma–Aldrich. For MIP synthesis, methacrylic acid (MAA) (Merck; Darmstadt, Germany) was used as a monomer, ethylene glycol dimethacrylate (EGDMA) (Xiya Reagent (Chengdu, China) was used as a cross linker, methanol (J.T. Baker, Phillipsburg, NJ, USA) was used as a porogenic solvent, glacial acetic acid (Brazil) was used for the removal of the template from MIP, 2,2′-azobis(isobutyronitrile) (AIBN) (Shanghai, China) was used as an initiator and Basic Blue 3 (Sigma Aldrich, Darmstadt, Germany) was used as a template molecule. A selectivity study of MIP for BB3 was carried out with MB, BB41, safranin and Thioflavin T. The water was deionized by the Milli-Q system. The chemical structures of the dyes used as given in Figure 1.

### 2.2. Characterization of Molecularly Imprinting Polymer and Non Imprinting Polymer

The structure of the polymer was analyzed using the Fourier-transform infrared spectroscopy (FTIR) Vertex 70 (Bruker, Billerica, MA USA) technique with a DLaTGS detector and a He Ne laser in the range of 4000–500 cm^−1^. The polymer size and morphology were determined using scanning electron microscopy (JSM-IT500, Zaventem, Belgium), and the surface area was determined by multipoint Brunauer–Emmett–Teller from the isotherm of nitrogen adsorption (ASAP 2010, Norcross, GA, USA). A total of 25 mg of dry sample was inserted in a sample holder before the measurement with the removal of N_2_ at 75 °C for 5 h using micrometrics at liquid nitrogen temperature. The absorbance measurements were carried out by a UV-Visible 1800 spectrophotometer using a quartz cuvette (Shimadzu, Kyoto, Japan).

### 2.3. Preparation of MIP and NIP

Mostly, the processes for MIP chemical synthesis involve noncovalent preassembly of the template functional monomer in solution followed by bulk polymerization [15]. In the current work, 2 mmol of the functional monomer (MAA) and 0.2 mmol of the template (BB3) were initially dissolved in 10 mL of methanol in a glass tube and then stirred for 4 min. After mixing, the solution was allowed to rest for 2 h. Then, 5 mmol of the cross liker (EGDMA) and 0.025 g of the initiator (AIBN) were added. The oxygen gas was removed from the mixture by nitrogen purging for 10 min. The sealed test tube was then placed in a water bath at 65 °C for 14 h. The polymer was grounded by mortar and pestle and then added into soxlet for washing with 3 rounds of methanol/acetic acid (7:3, *v*/*v*) and 3 rounds of methanol for the complete extraction of analyte. The same procedure as that described above was applied for non-molecularly imprinting polymer (NIP) synthesis, but without the template addition.

### 2.4. Binding Adsorption Analysis

The binding study of MIP was achieved using 20 mL vials containing 25 mg of MIP/NIP and 10 mL of 100 mg/L dye and adjusting parameters such as mass, concentration, pH and time. The mixture was then kept for 30 min on a shaker followed by centrifugation at 14,000 rpm and then filtration of the supernatant over 0.45 μm membrane (PTFE) before UV/V is spectrophotometric analysis. For the binding adsorption capacity, the following equation was used.
(1)$$Q=\frac{\left(C_{0}-C_{e}\right)}{V}m$$

*C*_0_ (mg·L^−1^) is the initial dye concentration, *C_e_* (mg·L^−1^) is the equilibrium dye concentration, Q (mg·g^−1^) is the experimental adsorption quantity, *V* (L) is the volume of solution and *m* (g) is the mass of MIP and NIP.

### 2.5. Selectivity Study

In order to check the formation of selective cavities in polymers, competitive adsorption analysis was carried out in the presence of those molecules that are similar as well as different in structure to BB3. In this regard different dyes such as Methylene blue (MB), Safranin, Basic blue 41 (BB41) and Thioflavin T were used. Each compound’s selectivity recognition experiments were carried out by dissolving 25 mg of MIP/NIP in a 10 mL solution (pH 11) containing 100 mg/L of each dye and equilibrated for 25 min. The concentration of these dyes in the supernatant was measured with a UV-Vis spectrophotometer.

### 2.6. Application in Real Water

To determine the application of MIPs for the preconcentration of reactive dyes in water, different samples were spiked with known amounts of dye concentration (100 mg/L) and subjected to the optimized extraction process. The extraction power of MIP seems to be affected in tap water and river water for BB3 dye, because dissolved organic and in-organic matter in the water samples played a role in slightly reducing the selective efficiency of MIP.

## 3. Results and Discussion

### 3.1. Choice of the Materials

Typically, stable and high affinity MIP synthesis requires the following reagents:

A high nominal level of cross linker is used for the template site preservation. A porogenic solvent and one or more functional monomer is needed to form a stable complex with the target molecule [16].

### 3.2. Characterization

#### 3.2.1. Characterization of Synthesized Polymer by SEM

With the help of scanning electron microscopy (SEM), the size information and geometry of the MIP can be obtained. Different research papers report the use of SEM to investigate MIP particles [17]. Therefore, SEM micrographs of the synthesized MIP and NIP were taken at different resolutions, as shown in Figure 2. The SEM images of both the MIP and NIP are not much different from each other. An overall assumption could be made from the general morphological architecture of both the MIP and NIP images for better general adsorption. The surface morphological appearance of all the SEM images is in favor of good adsorption ability in general. The SEM images are not very informative regarding the selective nature of MIPs, but they could be a good source for the general adsorption point of view [18].

#### 3.2.2. Characterization of Polymer by FTIR

In fact, FTIR analysis can reveal a lot about the nature of the surface functional groups that are formed on a polymer surface. The FTIR spectra of the MIP and NIPs within the range of 4000–500 cm^−1^ are shown in Figure 3. Both the MIPs and NIPs were synthesized from the same type of materials such as the template, monomer, initiator and cross linker; therefore, the spectra of the MIPs and NIPs show similarity [19]. The peaks around ~3400 cm^−1^ and ~2900 cm^−1^ show the presence of OH and CH bending, respectively, while the stretching at around~1700 cm^−1^ is due to the presence of C=O [20]. Therefore, the MIPs show a clear bend of the ester group at 1720 cm^−1^ and 1256 cm^−1^ due to the carbonyl C=O and C-O stretching. Additionally, the peaks at ~1400 cm^−1^ and ~1300 cm^−1^ are due to the presence of -CH_2_, and -CH_3_, respectively, while the strong peaks at 1159 cm^−1^ and 1046 cm^−1^ are due to the presence of the C-O functional group [21]. The FT-IR analysis of the MIPs and NIPs show that the MAA and EGDMA C=C double bond signal at 1637 cm^−1^ was absent, indicating that the polymer is successfully synthesized.

#### 3.2.3. Analysis by Brunauer–Emmett–Teller

The BET study determines the porosity and specific area of the selected BB3 molecularly imprinted polymers. The MIPs showed a specific surface area of 245.321 (m^2^/g), a pore volume of 0.078 (cc/g) and a pore radius of 14.492 (Å), while the NIPs have a 34.072 (m^2^/g), 0.0098 (cc/g) and 2.145 (Å) surface area, pore volume and pore radius, respectively (Table 1). The surface area of the NIP is almost seven times smaller than that of the MIP. The larger surface area of the MIP compared to the NIP indicates that specific cavities are formed for the recognition of BB3. The porosity of a particle refers to the volume ratio of the open pore to the total volume [22]. Previous research works show that the greater adsorption capacity of MIPs compared to NIPs is due to the higher surface area of MIPs as compared to NIPs. Additionally, MIPs have a higher total volume than NIPs due to their greater load capacity [23]. The larger pores of the MIP show that the structure of the NIP is more compact. The surface area and porosity of the MIP is mostly affected by the template in polymerization, and the average diameter of the pores for the MIP and NIP falls in the range of 2–50 nm, showing that both polymers are mesoporous [24]. A BET plot of the MIP and BJH plots for the MIP and NIP is shown in Figure 4.

### 3.3. Effect of Adsorbent Dose and pH

For the optimization of BB3 dye adsorption, the dose of the MIP was used in the range of (0.005–0.050 g) and 10 mL of 100 mg/L dye concentration. Figure 5 shows a linear increase in the removal of dye with an increase in the weight of the MIP and NIP from 0.0025 to 0.025 g due to the fact that the number of active sites of MIP/NIP increased. Further increase in the dose of MIP/NIP showed no effect on the uptake of the dye. The mechanism of dye adsorption depends on the degree of interaction with the adsorbent (MIP and NIP) surface and the protonation or deprotonation of dye, which may change adsorption efficiency [25]. So, for further studies, 0.025 g of polymer was used, and the pH of the test samples was varied in a range of 1–13, keeping the other parameters constant. It was observed that, at low pH, the adsorption is small. The adsorption capacity increases with increasing pH, and maximum adsorption was achieved at pH 11. So, this pH was selected as an optimum pH and used for further studies. With the change in pH, the surface charge varies. At lower pH, the surface charge is positive, whereas at higher pH, it becomes negative. Since the subject dye BB3 is a cationic dye, at higher pH, there will be strong interaction between the negative surface charge and the cationic BB3 dye. Due to this strong interaction, the optimum adsorption took place at pH 11. At higher pH, all sites became covered with negative charge; therefore, no further adsorption of BB3 dye took place. The results are shown in Figure 5.

### 3.4. Contact Time Studies as a Function of Temperature

Chemical kinetics is necessary to obtain and to find information about the binding mechanism and rate controlling process. The effect of contact time was examined as a function of temperature. This experiment was achieved using a fixed amount of polymer, 25 mg, with 10 mL of 100 mg/L dye solution at the optimized condition of pH, while varying the time from 1–35 min and the temperature from 283–313 K. The maximum dye adsorption was observed at 25 min, and then it remained constant. With the increase in temperature, a small change in the uptake of dye was observed, as shown in Figure 6.

### 3.5. Kinetic Models

To study the mechanism of dye adsorption on the adsorbent (MIP/NIP), we used three kinetics models: pseudo-first order, pseudo-second order and the Weber and Morris intraparticle diffusion model.

#### 3.5.1. Pseudo 1st t Order Kinetics

The pseudo-first kinetic model gives information about the rate of occupation of adsorption sites, which is proportional to unoccupied sites and is represented in Equation (2) [26].
(2)$$\log\;(q_{e}-q_{t})=\log\;q_{e}-K_{1}\;\frac{t}{2.303}$$

The values of q_t_ and q_e_ are the amount of BB3 dye (mg g^−1^) adsorbed at time t (min) and at equilibrium, respectively. *K*_1_ (min^−1^) is the pseudo-first order constant and is calculated by plotting log (q_e_ − q_t_) against “t”. The pseudo-first order kinetic model at different temperatures (283 K, 298 K, 313 K) is shown in Figure 7, and different parameters are given in Table 2. As the Q_e_ (cal) and Q_e_ (exp) do not match with each other, and the values of correlation coefficient are less than 0.99, we came to the conclusion that the adsorption of BB3 on MIP did not follow pseudo-first order kinetics.

#### 3.5.2. Pseudo 2nd Order Kinetic

The adsorption process can also be described by the pseudo-second order kinetic using the following Equation (3) [27].
(3)$$\frac{t}{q_{t}}=\frac{1}{k_{2}{{\;q}_{e}}^{2}}+\frac{t}{q_{e}}$$

In the above equation, k_2_ (mg g^−1^ min^−1^) is the constant of the pseudo-second order, which is calculated from the plot of t/q_t_ versus “t”. For the adsorption of BB3 dye on MIP, the pseudo-second order model was applied at different temperatures, as shown in Figure 7, while the different parameters calculated from this plot are given in Table 2. The R^2^ value for second-order kinetics is higher as compared to pseudo-first order kinetics, and a close resemblance between the Qe (cal) and Qe (exp) was found. So, we can conclude that the adsorption data of the pseudo second-order model are the best fit. These kinetic results show that the adsorption of BB3 dye depends on adsorbents as well as adsorbates.

#### 3.5.3. Intraparticle Diffusion Model

The kinetics data were also evaluated with the help of the intraparticle diffusion model proposed by Morris and Weber, which is given in Equation (4) [28].
(4)$$q_{t}=k_{\mathrm{id}}\;t^{1/2}+C$$
where C is the intercept which indicates the thickness of the boundary and k_id_ is the rate constant of intraparticle diffusion. According to this equation, a plot of qt versus t^1/2^ would be linear if the adsorption follows the process of the intraparticle diffusion model. Figure 7 shows a linear plot of the intraparticle diffusion for the adsorption of BB-3 on MIP. Multi linearity shows that the process of adsorption occurs in three steps. The first displays the diffusion of dye to the external surface of the adsorbent from the solution. The diffusion of the adsorbate from the external surface into the adsorbent pore is described by the second step. The final step is the adsorption of the adsorbate (BB-3) on the internal surface of the pore. However, at each temperature, the line fails to pass from the origin, which is due to the variance in the rate of mass transfer of initial and final. Furthermore, such a straight-line deviation from the origin reveals that the pore diffusion is not a sole rate control step [29]. Comparing the value of “C” and the rate constant indicates that the intraparticle diffusion is not the only rate limiting step. The adsorption of BB-3 on MIP is a complex phenomenon and is controlled by the surface sorption and intraparticle diffusion.

### 3.6. Adsorption Isotherms

The equilibrium adsorption isotherm study was carried out with varying dye concentrations (25, 50, 100, 150, 200 mg/L) to determine the efficiency of the prepared polymer, as shown in Figure 8. The figure shows that the isotherm rises abruptly at first, indicating that there are plenty of readily available sites for adsorption. When the adsorbent (MIP) becomes saturated after equilibration, a plateau is reached, indicating that no more sites are accessible for further adsorption. The experiments were conducted at 283 K, 298 K and 313 K to better realize the effect of temperature on dye adsorption. When the adsorption isotherms are compared, it is clear that adsorption increases as temperature rises, indicating that the process is endothermic.

The adsorption isotherm provides mechanism information of the adsorption process. Adsorption isotherms are used to assign the adsorption system, as they reveal the mechanism of adsorption. Generally, the Langmuir isotherm defines the monolayer formation and the non-covalent behavior of MIP, as described by the Freundlich model [30]. Therefore, the adsorption data were evaluated using the Langmuir and Freundlich isotherms.

#### 3.6.1. Langmuir Model

The Langmuir model defines a uniform and homogenous surface of the adsorbate that forms a monolayer. This model is based on a hypothesis: adsorption occurs at specific homogeneous sites within the body of the adsorbent. There is no contact between the adsorbed species and the adsorbent. The Langmuir model is described by Equation (5) [31].
(5)$$\frac{C_{e}}{Q_{e}}=\frac{1}{K_{L}Q_{m}}+\frac{C_{e}}{Q_{m}}$$
where *C_e_* (mg·L^−1^) is the liquid phase equilibrium concentration of the dye, *Q_m_* (mg·g^−1^) is the maximum adsorption capacity of the adsorbent, *K_L_* (L·mg^−1^) is the amount of dye adsorbed, the energy or the net enthalpy of adsorption and *Q_e_* (mg·g^−1^) is the quantity of dye adsorbed. The relationship between *C_e_*/q_e_ and *C_e_* must be linear, with a slope of 1/q_m_ and an intercept of 1/(q_m_
*K_L_*). The values of *K_L_*, *Q_m_* and R^2^ obtained from the curve are given in Table 3. The maximum adsorption capacities (Qmax) for MIP and NIP were 75.13 mg·g^−1^ and 7 mg·g^−1^, respectively. In Table 3, the Langmuir model provides an R^2^ value that best fits to the experimental value.

#### 3.6.2. Freundlich Model

The Freundlich isotherm is used to determine the adsorption properties of multilayer and heterogeneous surfaces with unequal adsorption sites and unusually available adsorption energies. The Freundlich isotherm model equation is given below [32].
(6)$$\ln\;q_{e}=\ln\;K_{f}+\frac{1}{n}\;\ln C_{e}$$
where *C_e_* (mg·L^−1^) is the liquid phase concentration at equilibrium, q_e_ mg·g^−1^ is the adsorb amount of dye, *K_f_* (mg·g^−1^) is taken as a relative indicator of the adsorption capacity and 1/*n* is the heterogeneity factor of the surface, which shows the adsorption nature. The 1/*n* value should be smaller than 1 for favorable adsorption, while for unfavorable adsorption, the value is greater than 1 and shows weak bond adsorption [33]. The Freundlich model is shown in Figure 9, and different parameters are given in Table 3.

### 3.7. Thermodynamic Study

Thermodynamic studies were undertaken to explore the adsorption mechanism feasibility of dye (BB-3) onto the MIP. In this study, a very significant decision has to be ended whether the process is spontaneous or not. Many thermodynamic parameters such as standard free energy (ΔG°), enthalpy (ΔH°) and entropy (ΔS°) were calculated using the following equation [34]:(7)$$\log K_{c}=\frac{\Delta S{}^{\circ}}{2.303R}-\frac{\Delta H{}^{\circ}}{2.303\mathrm{RT}}$$
(8)ΔG°=ΔH°−TΔS°
where T (K) is the absolute temperature, R (8.314 J·mol^−1^ K^−1^) is the universal gas constant and *K_c_* (L·g^−1^) is thermodynamic equilibrium constant described by q_e_/*C_e_*. The values of ΔS° and ΔH° were calculated from the intercept and the slope of a plot of log *K_c_* versus 1/T. Different thermodynamic parameters were studied at the different temperatures given in Table 4, and the Van’t Hoff plot is shown in Figure 10. At all temperatures, the value of ΔG° was found to be negative, which shows that the adsorption of dye (BB-3) onto the MIP is spontaneous [35]. By increasing the temperature, the value of ΔG° decreased, which indicates that high temperature facilitates the adsorption of dye (BB3) on MIP. The positive value of ΔH° specifies that this adsorption is endothermic, because increasing the temperature leads the rate of adsorbate diffusion to also increase on the adsorbent (external and internal surface). The positive value of ΔS° indicates that disorderedness increased during adsorption (solution–solid interface) [36,37].

### 3.8. Selectivity Study

For the confirmation of selective cavities formation in a polymer, competitive adsorption analysis was carried out in the presence of those dyes which are similar as well different in structure to the template Basic Blue 3 (BB-3), such as Basic Blue 41 (BB-41), Thioflavin T, Methylene blue (MB) and Safranin, whose structures are shown in Figure 11. The MIP was selective for the specific template (BB-3) as compared to the NIP. This study was performed by adding 100 mg/L of BB-3 dye and the interfering dyes to each Erlenmeyer flask at optimum conditions, and the mixture was stirred in a thermostat shaker at 125 rpm. At predetermined times, the samples were removed and the absorbance was measured by the UV-vis spectrophotometer. The amount of dye adsorbed on the polymer was calculated by substracting the final dye conentration from the initial dye added to a mixture. For BB-3, the MIP is more selective at 96% as compared to other dyes whose adsorption was between 5% and 41%. So, it confirms that the MIP cavities are selective only for the template (BB-3) molecule.

### 3.9. Imprinting Factor and Distribution Ratio

The imprinting factor describes the interaction and strength of the template molecule (BB3) toward the polymer (MIP/NIP). It indicates the recognition properties of the MIP and NIP toward specific analyte. The imprinting factor (IF) was calculated for the molecularly imprinting polymer using Equation (9).
(9)$$\mathrm{IF}=\frac{Q_{\mathrm{MIP}}}{Q_{\mathrm{NIP}}}$$
where Q_MIPs_ is the adsorption capacity of the MIPs for the dye (BB3) and Q_NIPs_ is the adsorption capacity of the NIPs for BB3. While, for the determination of the distribution ratio, the following equation was used.
(10)$$K_{d}=\frac{\left(C_{i}-C_{f}\right)}{C_{f}\;m}V$$
where K_d_ (L/g) is the distribution coeffecient, Ci is the initial dye concentration, C_f_ is the final dye concentarion, *V* is the volume used and “m” is the mass of the polymer (MIP/NIP). The different adsorption parameters estimated from various models are summurized in Table 5.

### 3.10. Real Sample Application

This study was undertaken to determine the application of MIPs for the preconcentration of reactive dyes in water. Molecularly imprinting polymers (MIP) were taken as a solid phase extracting material to evaluate their efficiency for the extraction of BB3 dye from environmental samples. The samples were spiked with known amounts of dye concentration (100 mg/L) and subjected to the optimized extraction process. It was found that the removal efficiency of dye (BB3) from the environmental samples was between 60–80%. The extraction power of MIP seems to be affected in tap water and river water for the extraction of BB3 dye. The dissolved organic and inorganic matter in the water samples played a role in slightly reducing the selective efficiency of the MIPs. Moreover, MIPs offer high selectivity towards reactive dyes, which is not present in expensive adsorbents such as activated carbon, biological treatment, etc. The results of BB3 dye extraction from different water samples are summarized in Table 6 whereas Table 7 summarizes the adsorption capacities of different adsorbents for the mentioned dye.

## 4. Conclusions

The current study focuses on the use of synthesized polymers (MIP) for the removal of a specific analyte (BB-3) from different water samples under optimized conditions. About 96.6% removal of BB-3 was achieved at 313 K, which indicates high selectivity toward the specific analyte as compared to other dyes such as safranin, Basic Blue 41 (BB-41), Thioflavin T (TT) and Methylene blue (MB), with 25%, 21%, 9.3% and 4.7%, respectively. Kinetic and isotherm studies showed that the adsorption of BB-3 on the polymer followed second order kinetics and the Langmuir model. The linear plot of qt vs. t^1/2^ in the intraparticle model revealed the surface sorption and intraparticle diffusion. In a thermodynamic study, the negative values of ΔG°, ΔH° and ΔS° demonstrate that the spontaneity, endothermic nature and disorder increased during the adsorption of BB-3 on the MIP, respectively. The MIP proved to be a workable material for the extraction and concentration of BB3 from effluent, as MIP can be easily removed from the media and recovered using centrifugation methods, allowing it to be reused without considerable activity loss.

## Figures and Tables

**Figure 1 molecules-27-03276-f001:**
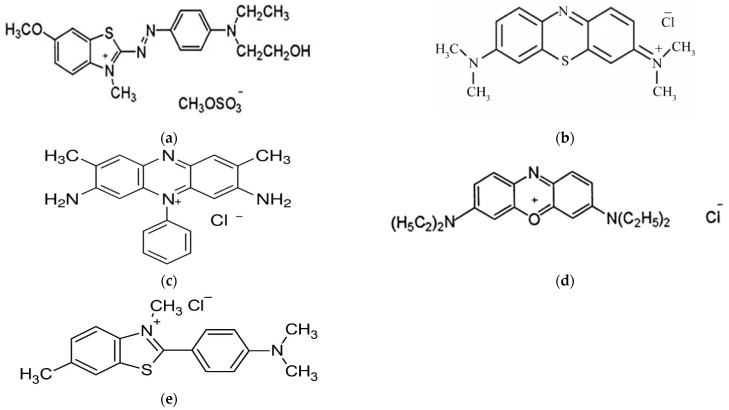
Structure of the different dyes: (**a**) Basic blue 41 (BB41), (**b**) Methylene blue (MB), (**c**) Safranin, (**d**) Basic blue 3, (**e**) Thioflavin T.

**Figure 2 molecules-27-03276-f002:**
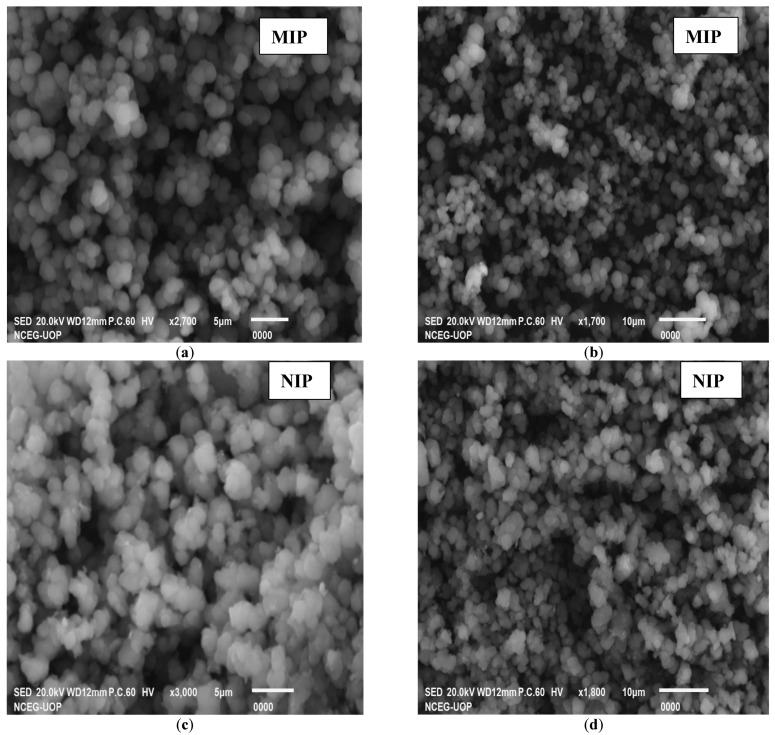
SEM images at different resolutions of the MIP (**a**,**b**) and NIP (**c**,**d**).

**Figure 3 molecules-27-03276-f003:**
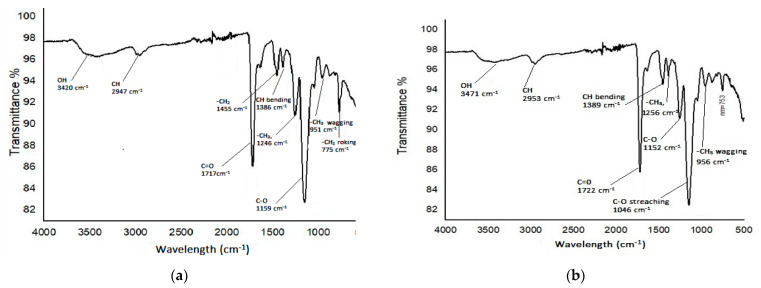
FTIR spectra of (**a**) MIP (**b**) NIP.

**Figure 4 molecules-27-03276-f004:**
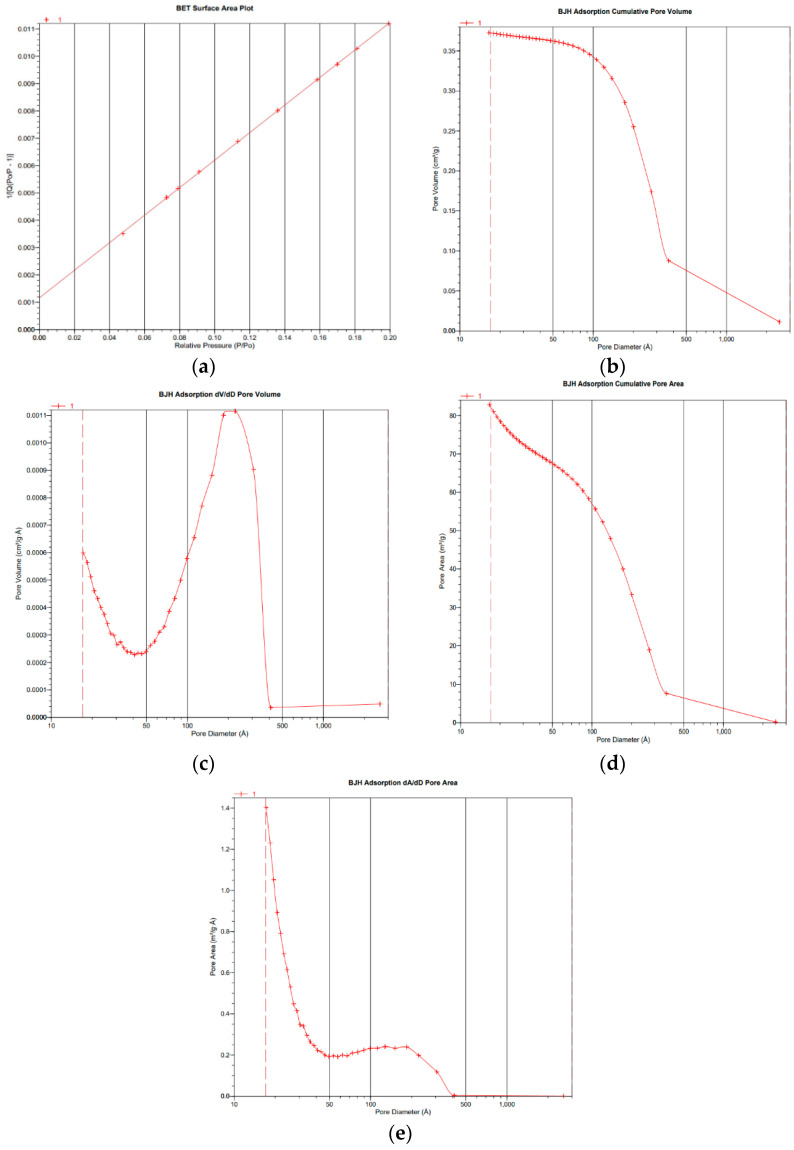
BET and BJH adsorption curves for MIP (**a**–**c**) and NIP (**d**,**e**).

**Figure 5 molecules-27-03276-f005:**
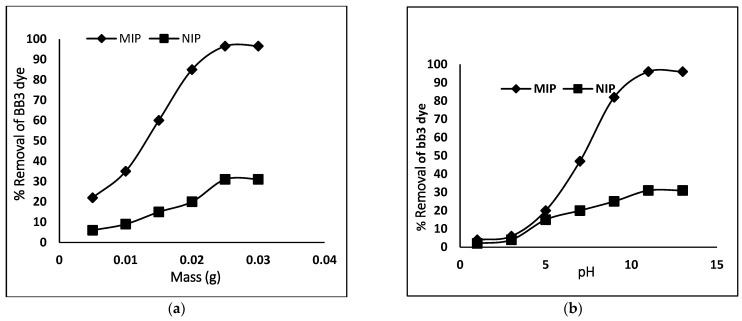
Removal efficiency of MIP/NIP for BB3 dye; (**a**) adsorbent dose (**b**) pH.

**Figure 6 molecules-27-03276-f006:**
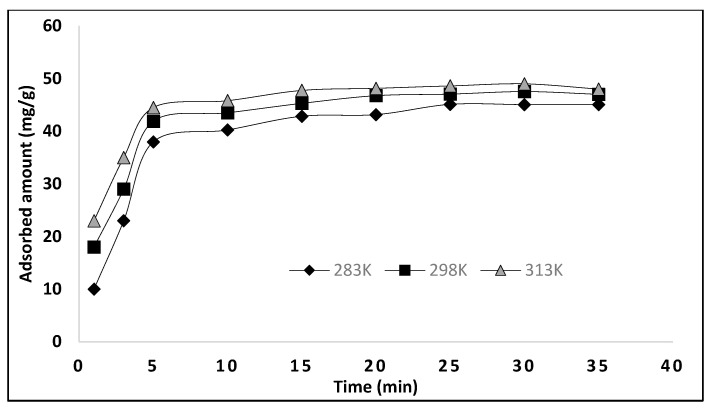
Adsorption kinetics for BB3 dye adsorption on MIP. Mass of polymer 25 mg; V = 10 mL; pH; 11; [BB3] = 100 mg/L.

**Figure 7 molecules-27-03276-f007:**
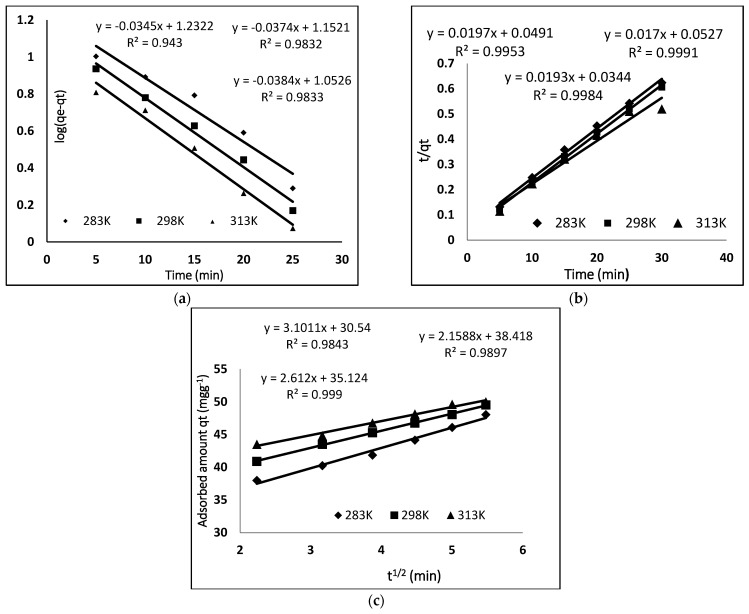
Different kinetic models for the adsorption of BB-3 on MIP; (**a**) pseudo-first order kinetic (**b**) pseudo-second order kinetic (**c**) intra particle diffusion model.

**Figure 8 molecules-27-03276-f008:**
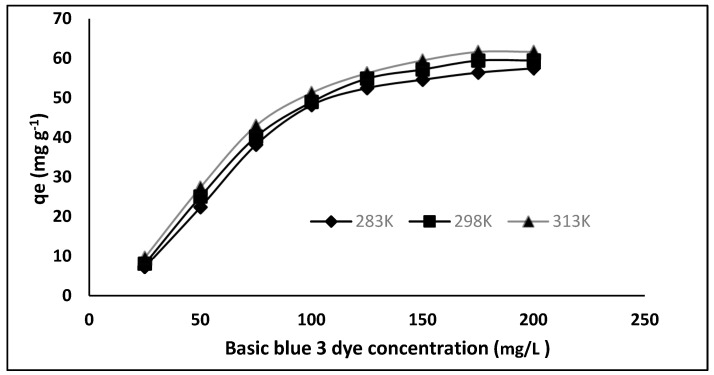
Adsorption isotherms of Basic Blue 3 dye adsorption on MIP adsorbent at different temperatures.

**Figure 9 molecules-27-03276-f009:**
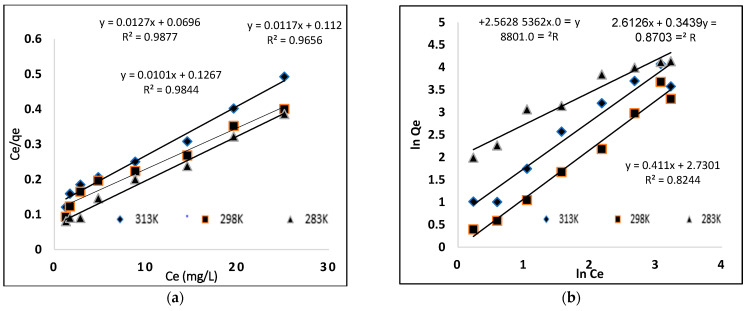
Isotherm models (**a**) Langmuir (**b**) Freundlich.

**Figure 10 molecules-27-03276-f010:**
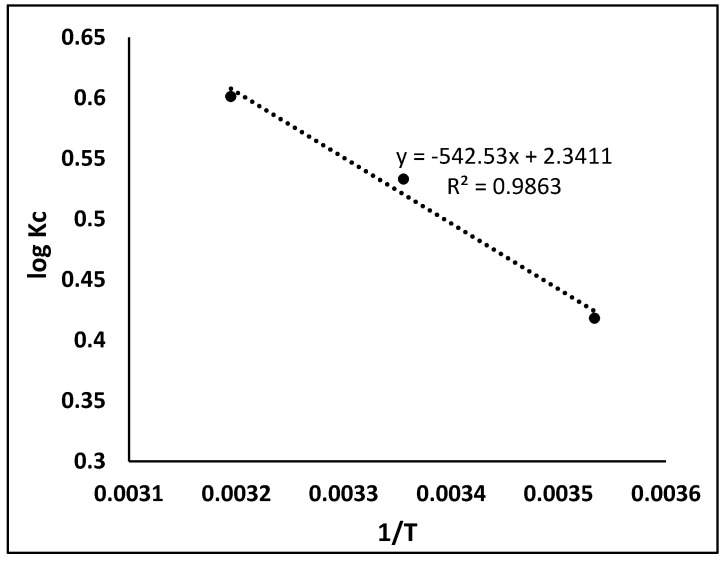
Van’t Hoff plotting for MIP.

**Figure 11 molecules-27-03276-f011:**
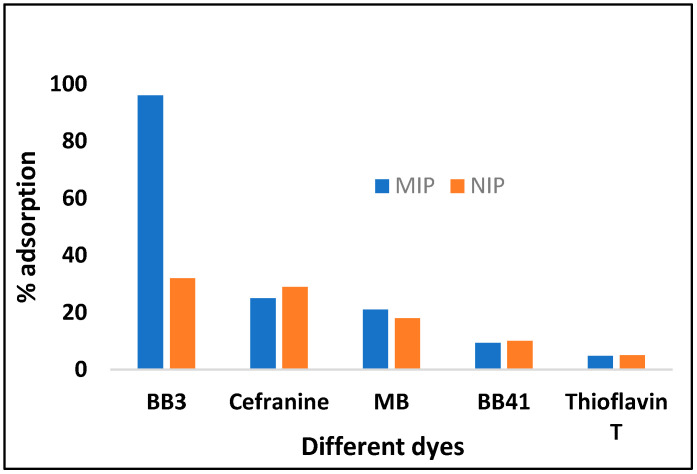
Effect of different dyes, showing the selectivity behavior of the MIP towards the adsorption of BB3 as compared to the NIP.

**Table 1 molecules-27-03276-t001:** BET analysis of MIP and NIP.

Polymers	Specific Surface Area (m^2^/g)	Pore Volume (cc/g)	Pore Radius (Å)
MIP	245.321	0.078	14.492
NIP	34.072	0.0098	2.145

**Table 2 molecules-27-03276-t002:** Different kinetic models’ parameters for the adsorption of BB3 on MIP.

Parameter	283 K	298 K	313 K
**Pseudo 1st order kinetic**			
K_1_	0.0794	0.0861	0.0951
*Q_e_* (cal)	17.068	14.197	11.287
*Q_e_* (exp)	48.049	49.528	49.996
R^2^	0.943	0.9832	0.9833
**Pseudo 2nd second kinetic**			
K_2_	0.0079	0.0108	0.0146
*Q_e_* (cal)	50.761	51.813	51.54
*Q_e_* (exp)	48.049	49.528	49.996
R^2^	0.9953	0.9984	0.9991
**Intraparticle diffusion**			
K_id_ (mg/g min ^−1/2^)	3.09	2.61	2.05
C	30.54	35.12	38.72
R^2^	0.9843	0.999	0.9897

**Table 3 molecules-27-03276-t003:** Different parameters of the Langmuir and Freundlich models for the adsorption of BB3 on MIP.

313 K	298 K	283 K	Parameters
**Langmuir isotherm model**			
91.743	83.33	78.125	*Q_m_* (mg·g^−1^)
0.1372	0.1872	0.1626	*K_L_*
0.9737	0.9837	0.9792	R^2^
**Freundlich**			
15.33	13.63	12.96	*K_f_* (mg·g^−1^) (L mg·g^−1^)
0.4152	0.3439	0.5362	1/n
0.8211	0.8703	0.8801	R^2^

**Table 4 molecules-27-03276-t004:** Thermodynamic parameters for the removal of BB3 on MIP.

Temperature	ΔG° KJ mol^−1^	ΔH° KJ mol^−1^	ΔS° J mol^−1^ K^−1^
283 K	−2057	10.38	44.82
298 K	−2735		
313 K	−3595		

**Table 5 molecules-27-03276-t005:** Different adsorption parameters for dyes by the polymer (MIP/NIP) at 283 K.

Dyes	% Removal		Adsorption Capacity Q (mg·g^−1^)		Distribution Coefficient Kd (L·g^−1^)		Imprinting Factor IF $\propto =\frac{Q_{MIP}}{Q_{NIP}}$	Selectivity $S=\frac{I^{BB3}}{I^{\;interfering}}$
	MIP	NIP	MIP	NIP	MIP	NIP		
BB3	96	31	78.4	7.0	0.48	0.07	10.73	-
Safranin	25	29	5.34	6.4	0.3	0.27	0.83	6.18
MB	21	18	4.03	3.26	0.02	0.03	1.23	9.66
BB41	9.3	10	1.75	1.87	0.19	0.05	0.93	7.37
Thioflavin T	4.76	5.0	2.0	2.11	0.20	0.21	0.94	7.29

**Table 6 molecules-27-03276-t006:** MIP efficiency for the removal of BB3 from different environmental samples.

Samples	Amount of BB3 Added (mg/L)	Amount of BB3 Found (mg/L)	%Recovery ±SD
Distilled water	100	96.2	96.2 ± 0.9
River water	100	61.1	31.1 ± 0.1
Tap water	100	82.09	32.09 ± 1.2

**Table 7 molecules-27-03276-t007:** Comparison of the adsorption capacities of different adsorbents to BB3 dye.

Adsorbents	Adsorption Capacity (mg·g^−1^)	Ref
Chitosan-based	166.5	[38]
Amberlite XAD 1180	66.5	[39]
Aleppo pine-tree sawdust	65.4	[40]
Pineapple stem	58.9	[41]
Durian husk	49.5	[42]
Acrylic resin	46.95	[43]
Peat	41.00	[44]
Risk Hull	13.41	[45]
MIP	75.125	Present study

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
