# Peer review of "Selective Removal of the Emerging Dye Basic Blue 3 via Molecularly Imprinting Technique"

_molecules, 2022, doi:10.3390/molecules27103276_

Round 1

Reviewer 1 Report

In this manuscript, the authors have investigated the selective removal of dyes through molecular imprinting polymers (MIP). Mainly, the authors have not followed the journals guidelines while writing the manuscript. Several typos have appeared in the whole manuscript and the graphs and Fig. captions are confused. The abbreviations are not in proper order and confusing. The scientific invention in this manuscript is well fit for publication in Molecules however, it will be keenly rectified the typos. The manuscript contains useful and valuable information for the removal of dye samples using MIP.

As far as I am concerned, the authors should address the following points before publication.

Q1. What is MIP? Authors have not abbreviated anywhere in the manuscript, however, they have used the word MIP regularly both in abstract and body text. I recommend authors should provide the expansion of MIP.

 Q2. Since FT-IR analysis is key for this research, hence, the authors should provide a color image of Figure 3 with significant difference/enhancement of peak area. It will be fair enough for the readers to understand, and otherwise it looks messy. And authors have to explain the exact peak to peak functional group identification for the MIP and NIPs in the FT-IR spectra

Q3. Sec. 3.2.3. It will be good to readers if authors will provide BET graphs of both MIP and NIP.

Q4. Authors should confirm is it Fig. 4 or Fig.1? In the Sec. 3.3. More importantly, the data points are not matching in the graphs of BB-3 dye against MIP and NIP? Rectify them.

Q5. The manuscript contains poor quality graphs, I strongly recommend to authors should replace with thick lines.Q6. I recommend for the publication with the above inputs.

Author Response

Question 1: What is MIP? Authors have not abbreviated anywhere in the manuscript; however, they have used the word MIP regularly both in abstract and body text. I recommend authors should provide the expansion of MIP.

Ans: Worthy reviewer, MIP is the abbreviation for Molecularly Imprinting Polymer and it has now been mentioned in Abstract as molecularly imprinting polymer (MIP).

Q 2: Since FT-IR analysis is key for this research, hence, the authors should provide a color image of Figure 3 with significant difference/enhancement of peak area. It will be fair enough for the readers to understand, and otherwise it looks messy. And authors have to explain the exact peak to peak functional group identification for the MIP and NIPs in the FT-IR spectra.

Ans: Resolution of FTIR spectra of MIP and NIP has been enhanced and important peaks have been mentioned in the Spectra as well as in text of manuscript.

Q 3: Sec. 3.2.3. It will be good to readers if authors will provide BET graphs of both MIP and NIP.

Ans: Due to the increasing number of plots, All the three plots of MIP i.e. BET surface area plot and BJH plot have been added in the manuscript and shown in figure 4 but for NIP only two graphs have been added in the manuscript.

Q4. Authors should confirm is it Fig. 4 or Fig.1? In the Sec. 3.3. More importantly, the data points are not matching in the graphs of BB-3 dye against MIP and NIP? Rectify them.

Ans: It is figure 5 and it has been corrected in manuscript as well. Figure 5 has also been corrected

Q 5: The manuscript contains poor quality graphs I strongly recommend to authors should replace with thick lines.

Ans: All the graphs have been replaced with thick lines.

Reviewer 2 Report

  1. Use the MIP and NIP abbreviation at first appearance.
  2. why pH 11 was selected for selectivity study for all dyes. 
  3. Improve the language of the article
  4. How the authors are so sure that MIP is successfully synthesized? 
  5. authors must use the proper scientific language eg. 3.1. Choice of the reagents. It is the materials.
  6. authors wrote in SEM that "the shape of MIP particles is more porous and uniform and also small in size as compared to NIP. The MIP surface area showed better-defined sites than NIP so that’s why the recognition ability of MIP towards analyte (BB3) is higher than NIP". I am not agreeing with the justification, pores is not seen, and the shape is not uniform. 
  7. correct the units. 
  8. why optimum adsorption was observed at pH 11, Justify? 
  9. I am not satisfied with the kinetic study plot because the authors mention that equilibrium was established within 25 min, but after 25 min authors reported only one point, authors must do at least 3-4  point data. I feel the equilibrium was established in 5 min, so the authors must do more experiments between 0- 5 min
  10. authors must the isotherm models at all temp.
  11. Write the novelty statement
  12. write the more justification for all sections
  13. Abstract and conclusion must be more informative. 

Author Response

Reviewer 2

Q 1: Use the MIP and NIP abbreviation at first appearance.

Ans: Worthy reviewer, the abbreviations have been added at first appearance in abstract section of the manuscript

Q 2: why pH 11 was selected for selectivity study for all dyes.

Ans: Optimization studies for maximum adsorption of basic blue 3 dye (BB3) were carried out at different pH and optimum pH for subject dye i.e.BB3 was found to be 11 with 96% adsorption. Therefore, to carry out the effect of interfering dyes, the competitive experiments for all other dyes in the presence of subject dye were carried out at optimum condition of pH for the same cavities that were made for the specific template (BB3).

Q 3: Improve the language of the article

Ans: The language has been improved

Q 4: How the authors are so sure that MIP is successfully synthesized?

Ans: As we know that MAA and EGDMA have C=C double bond in their structures and for the formation of polymer, the breakdown of this bond is necessary. In FTIR analysis, the peak for this C=C appears around 1600 cm-1 and in FTIR spectrum of molecularly imprinting polymer, the peak around 1600 cm-1  was absent thus confirming the successful polymerization through this functional group.

Q 5: authors must use the proper scientific language eg. 3.1. Choice of the reagents. It is the materials.

Ans: The correction has been made.

Q 6: authors wrote in SEM that "the shape of MIP particles is more porous and uniform and also small in size as compared to NIP. The MIP surface area showed better-defined sites than NIP so that’s why the recognition ability of MIP towards analyte (BB3) is higher than NIP". I am not agreeing with the justification, pores are not seen, and the shape is not uniform.

Ans: The SEM images of both MIP and NIP are not much different from each other.

An overall assumption could be made from the general morphological architecture of both MIP and NIP images for the better general adsorption. The surface morphological appearances of all the SEM images are in favor of good adsorption capability in general. The SEM images are not much informative regarding the selective nature of the MIPs but could be a good source from general adsorption point of view

Q 7: correct the units.

Ans: Units throughout the manuscript have been corrected.

Q 8: why optimum adsorption was observed at pH 11, Justify?

Ans: With change in pH the surface charge varies. At lower pH the surface charge is positive whereas at higher pH it becomes negative. Since the subject dye Basic Blue 3 (BB3) is a cationic dye therefore at higher pH, there will be strong interaction between negative surface charge and the cationic BB3 dye.  Due to this strong interaction optimum adsorption was observed at pH 11. After pH 11, all sites became covered with negative charge and no change in adsorption took place.

Q 9: I am not satisfied with the kinetic study plot because the authors mention that equilibrium was established within 25 min, but after 25 min authors reported only one point, authors must do at least 3-4 point data. I feel the equilibrium was established in 5 min, so the authors must do more experiments between 0- 5 min

Ans: The whole kinetic study was repeated to find out the exact equilibrium time and the results are shown in Figure 5. After performing a series of experiments at different times ranging from 1-35 minutes, it was found that equilibrium established after 25 minutes, after that only small change in adsorption was observed. Therefore 25 minutes was then selected as optimum time.

Q 10: authors must the isotherm models at all temp

Ans: As per your suggestion the isotherm models have been applied at three different temperatures i.e. 283 K. 293 K and 313 K and are given in figure 8. Different parameters calculated from these isotherms at different temperatures are given in table 3.

Q 11: Write the novelty statement

Ans: The Basic Blue 3 being a recently emerging dye on the market is used in different industries. The selective removal of the dye is critical owing to the presence of many toxic groups such as aromatic amines. Although the MIP is an already reported technique but in current study it has been employed for the first time using the Basic blue 3 as the template. The selective removal of the Basic blue 3 with high efficiency and improved throughput analysis is the novelty of current work in comparison to the previously reported literature about Basic Blue 3.

Q 12; write the more justification for all sections

Ans: More justification where possible have been added in different sections.

Q 13: Abstract and conclusion must be more informative.

Ans: More information have been added in abstract as well as in conclusion part of the manuscript.

Round 2

Reviewer 2 Report

acceptable